# Clinical validation of the Roche cobas HPV test on the Roche cobas 5800 system for the purpose of cervical screening

Nikita Mehta,[1] Marco Ho Ting Keung,[1] Eunice Pineda,[1] Elliott Lynn,[1] Dagnachew Fetene,[1] Alvin Lee,[1] Nicolas Hougardy,[2] Amelie Heinrichs,[2] Hiu Tat Mark Chan,[1,3] Marc Arbyn,[4] Marion Saville,[1,5] David Hawkes[1,6,7]

**ABSTRACT** This study assessed the relative clinical sensitivity and specificity, as well as reproducibility, for high-risk HPV types of the Roche cobas HPV test when processed using the Roche cobas 5800 system. The results from this study demonstrate that the cobas HPV test using the cobas 5800 system fulfils the Meijer criteria for use in population-based cervical screening. This clinical validation study also examines the clinical sensitivity and specificity based on partial genotyping, with separate detection of HPV16 and HPV18, compared with the Roche cobas 4800 HPV test, a second-generation standard comparator assay. The cobas HPV test has a relative clinical sensitivity of 1.000, when compared with the cobas 4800 HPV test to detect histologically confirmed CIN2+ lesions in woman aged 30 years or older, with a relative clinical specificity of 0.995. The general intra- and inter-laboratory agreement for the cobas HPV test on the cobas 5800 system for finding a HPV positive result were 99.1% and 99.6%, respectively.

**IMPORTANCE** This study demonstrates, for the first time, the clinical performance of the Roche cobas HPV test when processed using the new cobas 5800 system [cobas (5800)]. This study shows that the cobas (5800) demonstrates relative clinical sensitivity and specificity, when compared with a standard comparator HPV test, which meets the international HPV test validation requirements. Intra- and inter-laboratory reproducibility also fulfills these criteria. The current study demonstrates that the cobas (5800) can be used for primary HPV-based cervical screening on cervical specimens.

**KEYWORDS** HPV, clinical validation, diagnostic, cervical screening, cervical intraepithelial neoplasia

I n 2020, the World Health Organization (WHO) announced the strategy to accelerate the elimination of cervical cancer as a public health problem (1).The second pillar of this strategy is for countries to screen 70% of women by the age of 35 and again by the age of 45 with a high-performance test. Human papillomavirus (HPV) nucleic acid amplification testing (NAAT) meets the criterion of being a high-performance test. Since 2020, and especially since the end of the international measures in response to the COVID-19 pandemic, there has been increased interest in HPV-based cervical screening programs. In spite of over 260 commercially available tests in 2023 (2), there are currently only 20 clinically validated HPV assays (3, 4), and some of these are not commercially available, not suitable for the volumes associated with a screening program, or lack a cellularity control, as recommended in the guidelines from the International Papillomavirus Society (IPVS) (5) and in a draft version of the WHO HPV Target Product Profile (TPP) released for public consultation (6).

The requirements for clinical validation of HPV assays for use in cervical screening has often followed the criteria first published by an international consortium in 2009, the Meijer Criteria (7, 8). These criteria examine whether the candidate assay's clinical

Address correspondence to David Hawkes, dhawkes@vcs.org.au.

Nikita Mehta and Marco Ho Ting Keung contributed equally to this article. The author order was determined by who the initial Project Lead was.

VCS Pathology (employer of N.M., M.H.T.K., E.P., E.L., H.T.M.C., M.S., and D.H.), a division of ACPCC, has received research funding or gratis consumables to support laboratory-based research from the following commercial entities in the last 3 years: Abbott, AusDiagnostics, BD, Cepheid, Copan, Qiagen, Roche, Rovers, and Seegene. D.H. is an investigator on the Compass, MaRVE, and SCoPE2(VALHUDES) clinical trials. M.S. is a co-principal investigator (PI) of an investigator-initiated trial of HPV screening in Australia (Compass), which is conducted and funded by the Australian Center for the Prevention of Cervical Cancer (ACPCC), a government-funded health promotion charity. The ACPCC has previously received equipment and a funding contribution for the Compass trial from Roche Molecular Systems USA. M.S. is also a co-PIs on a major implementation program, "Elimination of Cervical Cancer in the Western Pacific," which receives support from the Minderoo Foundation and equipment donations from Cepheid Inc. M.A. was supported by the Horizon 2020 Framework Program for Research and Innovation of the European Commission, through the RISCC Network (Grant No. 847845). Sciensano, the employer of M. Arbyn received funding from the European Society of Gynecological Oncology (ESGO), German Guideline Program in Oncology (German Cancer Aid project), World Health Organization (Geneva, Switzerland, via Agreement for Performance of Work for guidelines on Screening and Treatment of Pre-Invasive Cervical Disease); the European Commission Initiative on Cervical Cancer (EC-CvC) and the VALGENT and VALHUDES projects. D.F., A.L., N.H., and A.H. have no conflicts of interest.

See the funding table on p. 7.

sensitivity and specificity for cervical intraepithelial neoplasia grade 2 or above (CIN2+) is non-inferior to a standard comparator HPV test. Reproducibility is also assessed with a cohort of at least 500 samples tested twice in the same laboratory (intra-laboratory) at the suggested interval of "several weeks later" (7) and then re-tested in an independent external laboratory (inter-laboratory). Originally, the Meijer criteria (7) identified two standard comparator HPV assays, the Digene Hybrid Capture 2 and GP5+/6+-PCR-EIA (9–11). However, second-generation standard comparator assays have been described by Arbyn and colleagues, which include the Roche cobas 4800, the Abbott Realtime, the BD Onclarity, and the Seegene Anyplex II HR HPV assays (4). The Roche cobas 4800 HPV test (cobas 4800) has been used as an alternative comparator test in two studies evaluating the cobas HPV test on the cobas 6800 system. The current validation utilized the cobas 4800 as the standard comparator assay.

The Roche cobas HPV test, which can be processed on the Roche cobas 5800, 6800, or 8800 systems, has only been clinically validated for sensitivity, specificity, or reproducibility on the cobas 6800 system prior to this study (12, 13). The current study [Meijer and Reference Validation Evaluation (MaRVE)] is assessing the clinical performance of the cobas HPV test when run on the cobas 5800 system [cobas (5800)] using cobas 4800 as a comparator, which currently is accepted as a second-generation standard comparator test (4).

The cobas HPV test detects 14 HPV types previously referred to as "high-risk" but more commonly separated into the 12 Group 1 (carcinogenic to humans) carcinogenic HPV types (14), HPV16, 18, 31, 33, 35, 39, 45, 51, 52, 56, 58, and 59, and two other HPV types 68, and 66, which are categorized as Group 2A and 2B carcinogenic HPV types, respectively. These 14 HPV types are included in the majority of the clinically validated HPV tests. The cobas HPV test also gives partial genotyping by separately identifying HPV16 and HPV18, with the results for the remaining 12 types reported in a combined "pool." The cobas HPV test also utilizes the human beta-globin gene target as an endogenous control for inhibition and cellularity. The cobas HPV test is being used in several national cervical screening programs, including Australia (15), New Zealand, and England.

## MATERIALS AND METHODS

*Study population:* Residual cervical specimens collected from women who participated in the HPV-based Australian National Cervical Screening Program (NCSP) in 2023. The Australian NCSP involves primary HPV-based screening for all people with a cervix, aged between 25 and 74, with a 5-year screening interval. Clinician-collected cervical specimens were collected as per standard practice and immediately resuspended into 20 mL of PreservCyt (a ThinPrep vial, Hologic, Bedford, MA, USA) at the point of collection. The study was approved by the Bellberry Human Research Ethics Committee (#2023–11-1356). All residual clinical specimens were obtained from women aged at least 30 years for our analyses following the international validation guidelines (7). Cervical specimens in the primary ThinPrep vial were processed using the Roche 480 pre-analytic instrument, which aliquots 1 mL of the specimen into a Roche cobas secondary tube, which can then be run on both the Roche cobas 4800 and cobas 5800 systems.

1. For clinical sensitivity, we identified a set of 63 cervical screening samples from women who subsequently had histologically confirmed CIN2+ and/or adenocarcinoma-*in situ* (AIS) referred to as "cases" hereafter. The median age of cases was 38 years (range 30 to 70 years). Only samples with a valid test result on cobas 4800 were included as cases. The cases comprised 23 CIN2, four CIN2/CIN3, 34 CIN3, one AIS/CIN3, and one AIS.

2. For clinical specificity, a total of 821 consecutive cervical screening samples meeting the study criteria from women who had no evidence of histologically confirmed CIN2+, AIS, or worse, were referred to as "controls." Absence of disease was not verified by colposcopy or biopsy. Only specimens from individuals who were routine participants in the Australian NCSP and who had both a previous routine HPV negative screening test (>57 and <84 months prior—the window for a routine screening interval), which was immediately (≥2 years) preceded by a negative routine cytology-based screening episode. There were 53 controls, which resulted in a HPV positivity rate of 6.46%. This is within the 95% CI range of 5.24%–8.88% for a cohort of 850 specimens according to the Australian National Cancer Screening Register at the time of the study in December 2023 (16). Specimens from women whose test was undertaken as a follow-up for a previous abnormality, a HPV detected test, or as "test-of-cure" after histologically confirmed and treated CIN2+/AIS were excluded. The median age of controls was 54 years (range 30 to 74 years). Only samples with a valid test result on cobas 4800 were included in the controls. For intra-laboratory reproducibility, we identified a set of 550 cervical samples, which were identified in the same manner as the specificity cohort described above, until there were 385 HPV not detected and 165 HPV detected (30%) cases on the cobas (5800). All 550 samples were tested twice at the Australian HPV Reference Laboratory, a division of the Australian Center for the Prevention of Cervical Cancer (ACPCC) laboratory using the cobas (5800), with an interval range of 34–43 days between the two testing events.

3. For inter-laboratory reproducibility, the samples were then tested at another laboratory (Vivalia, Arlon, Belgium) using the cobas (5800). There was an interval range of 58–77 days between the two testing events.

## Clinical performance & reproducibility assessment

cobas (5800) clinical accuracy to detect cervical intraepithelial neoplasia of grade 2 or worse (CIN2+) was assessed against the second-generation standard comparator (cobas 4800). Moreover, it was verified as to whether the clinical accuracy of cobas (5800) was non-inferior to that of cobas 4800, using the statistics of Tang for paired comparisons, accepting 0.90 (for relative sensitivity) and 0.98 (for relative specificity) as benchmarks. Practically, this means that the left bound of the 90% confidence intervals around the relative sensitivity or specificity is not lower than the respective benchmark (3, 7). To assess the cobas (5800) intra-laboratory reproducibility and inter-laboratory agreement, a lower confidence bound of ≥87% and a kappa value of at least 0.5 were used as benchmarks (7).

All statistical analyses were carried out in Stata 16.1 (College Station, TX, USA), and a $P$ value below 0.05 was considered significant.

## RESULTS

Clinical sensitivity and specificity for *CIN2+:* Of the 63 cases, 61 had HPV detected on the cobas (5800) HPV test, resulting in an absolute clinical sensitivity of 96.8% (95% CI: 92.5% to 100%). By comparison, 61 cases had HPV detected on the cobas 4800, resulting in the same absolute clinical sensitivity of 96.8% (95% CI: 92.5% to 100%, Table 1). The relative clinical sensitivity of the cobas (5800) HPV test was 1.000 (90% CI: 0.963 to 1.039) compared to the cobas 4800 and assessed to be non-inferior ($P < 0.0082$).

Absolute clinical sensitivity for CIN3+ for the cobas 4800 was 100% as all 40 CIN3+ cases had HPV detected. The cobas (5800) detected HPV in 39 CIN3 cases, which demonstrated an absolute analytical sensitivity of 97.5% (95% CI: 92.7% to 100%). The relative clinical sensitivity for CIN3+ was 0.975 (90% CI: 0.975 to 1.016) compared to the cobas 4800.

Two samples were discordant with one sample, a CIN2 case, for which HPV was not detected on the cobas 4800, but HPV Other was detected on the cobas (5800), with

**TABLE 1** HPV test findings among 884 cervical samples included in the MaRVE study, stratified by the presence or absence of histologically confirmed CIN2+ or CIN3+ lesions

| Clinical status | cobas (5800) HPV test result | cobas 4800 HPV test result | | Total |
|---|---|---|---|---|
| | | Negative | Positive | |
| Controls | Negative | 763 | 1 | 764 |
| (<CIN2) | Positive | 5 | 52 | 57 |
| | **Total** | **768** | **53** | **821** |
| Cases | Negative | 1 | 1 | 2 |
| (≥CIN2) | Positive | 1 | 60 | 61 |
| | **Total** | **2** | **61** | **63** |
| Cases | Negative | 0 | 1 | 1 |
| (≥CIN3) | Positive | 0 | 39 | 39 |
| | **Total** | **0** | **40** | **39** |

a cycle threshold (ct) value indicating a moderate positive result (> 6 cycle threshold stronger than the manufacturer's stated limit of detection (LOD) ct value ) (Table 2). The second discordant sample was a CIN3 case that had HPV Other detected (beyond LOD ct) on the cobas 4800 but no HPV detected on the cobas (5800).

Of the 821 controls, 764 were negative for HPV on the cobas (5800), resulting in an absolute clinical specificity of 93.1% (95% CI: 91.3% to 94.8%, Table 1). The absolute clinical specificity of cobas 4800 for CIN2+ was 93.5% (95% CI: 91.9% to 95.2%) (768/821 controls). Relative to the cobas 4800, the cobas (5800) had a specificity of 0.995 (90% CI: 0.99 to 1.00) and was assessed to be non-inferior ($P < 0.0024$) for histologically confirmed CIN2+ lesions in woman aged 30 years or older.

Of the five controls that had HPV detected on the cobas (5800) HPV test but were negative on the cobas 4800, HPV 16 was detected in four and HPV Other in the other specimen. All five results were at or beyond the LOD ct value. When the relative clinical sensitivity for CIN2+was assessed by partial genotyping by HPV type (HPV16, HPV18, or the group of 12 HPV types—"HPV Other"), there was relative clinical sensitivity for HPV16 of 1.125, HPV18 of 1.000. and HPV Other of 0.957 (Table 2).

*Intra-laboratory and inter-laboratory reproducibility:* The intra-laboratory reproducibility over time was 99.6% (548/550; 95% CI: 98.9% to 99.9%), with a kappa value of 0.991 (Table 3). The inter-laboratory reproducibility was 99.8% (549/550; 95% CI: 99.2%

**TABLE 2** Type-specific HPV detection using the cobas (5800) HPV test in relation to the cobas 4800 HPV test stratified by the presence or absence of histologically confirmed CIN2+ lesion

| Clinical status | cobas (5800) HPV test result | cobas 4800 HPV test result | | | | | | | Total |
|---|---|---|---|---|---|---|---|---|---|
| | | HPV negative | HPV16 | HPV18 | Other HPV | HPV16 & Other HPV | HPV18 & Other HPV | HPV16 & HPV18 & Other | |
| Controls | HPV negative | 763 | 0 | 0 | 1 | 0 | 0 | 0 | 764 |
| (<CIN2) | HPV16 | 4 | 4 | 0 | 0 | 0 | 0 | 0 | 8 |
| | HPV18 | 0 | 0 | 0 | 0 | 0 | 0 | 0 | 0 |
| | Other HPV | 1 | 0 | 0 | 45 | 0 | 0 | 0 | 46 |
| | HPV16 & Other HPV | 0 | 0 | 0 | 1 | 0 | 0 | 0 | 1 |
| | HPV18 & Other HPV | 0 | 0 | 0 | 0 | 0 | 2 | 0 | 2 |
| | **Total** | **768** | **4** | **0** | **47** | **0** | **2** | **0** | **821** |
| Cases | HPV negative | 1 | 0 | 0 | 1 | 0 | 0 | 0 | 2 |
| (CIN2+) | HPV16 | 0 | 11 | 0 | 0 | 0 | 0 | 0 | 11 |
| | HPV18 | 0 | 0 | 3 | 0 | 0 | 1 | 0 | 4 |
| | Other HPV | 1 | 0 | 0 | 37 | 0 | 0 | 0 | 38 |
| | HPV16 & Other HPV | 0 | 0 | 0 | 2 | 4 | 0 | 0 | 6 |
| | HPV18 & Other HPV | 0 | 0 | 0 | 0 | 0 | 1 | 0 | 1 |
| | HPV16 & HPV18 | 0 | 0 | 0 | 0 | 0 | 0 | 1 | 1 |
| | **Total** | **2** | **11** | **3** | **40** | **4** | **2** | **1** | **63** |

**TABLE 3** Intra-laboratory reproducibility over time of cobas (5800) HPV test for HPV detection

| ACPCC, First Test | ACPCC, Second Test | | |
|---|---|---|---|
| | Negative | Positive | Total |
| Negative | 383 | 2 | 385 |
| Positive | 0 | 165 | 165 |
| **Total** | **383** | **167** | **550** |

to 100%), with a kappa value of 0.996 for the first test and a reproducibility of 99.5% (547/550; 95% CI: 98.6% to 99.8%) and a kappa value of 0.987 for the second test (Table 4). Both the intra-laboratory reproducibility over time and the inter-laboratory agreement fulfilled the validation criteria.

*Discordant results:* Of the three discordant specimens, all tested negative in the first intra-laboratory test. Two specimens only detected HPV for the second test, one detected HPV16, and the other detected HPV Other. One specimen only detected HPV (Other) for the intra-laboratory (Vivalia) test. All three discordant specimens had results at or beyond the LOD ct values.

## DISCUSSION

In this study, the clinical performance of the cobas (5800) was compared to that of the validated second-generation standard comparator cobas 4800, in a cohort of screening participants aged 30 years or older. The relative clinical sensitivity and specificity for histologically confirmed CIN2+ of the cobas (5800) were non-inferior to those of the cobas 4800. The cobas (5800) also demonstrated near perfect levels of intra- and inter-laboratory reproducibility. The use of the cobas 4800 as the standard comparator test also allowed the comparison of partially genotyped results. The benefit of being able to undertake this comparison is relevant in a growing number of national cervical screening programs, including Australia, the Netherlands, and New Zealand, which use partial genotyping as part of the clinical management of women in whom HPV was detected. Cases that had HPV16 and/or HPV18 detected demonstrated 91.3% (21/23) concordance, with a relative clinical sensitivity of 1.095 between the cobas (5800) and cobas 4800 HPV tests. HPV16 and HPV18 also demonstrated high levels of concordance when assessed for intra- and inter-laboratory reproducibility (Table 5).

The strengths of this study are the use of a representative screening population recruited from participants in the second round of a primary HPV-based National Cervical Screening Program in Australia, the known screening history including previous HPV-based and cytology-based screening episodes, and the use of agreed published international methodological criteria for assessment.

The cobas (5800) is a useful addition as a clinically validated HPV test as it provides a more modern, versatile, and faster (2 hours 45 minutes to first result) option when compared with the cobas 4800 system (4 hours 45 minutes to first result), which was launched 15 years ago. The cobas 4800 system is more manual and requires more staff hands on time (estimated as 39 minutes per run of 96 samples). The cobas 5800 system

**TABLE 4** Inter-laboratory agreement of cobas (5800) HPV test for HPV detection

| ACPCC test results | Vivalia result | | Total |
|---|---|---|---|
| | Negative | Positive | |
| First Test | | | |
| Negative | 384 | 1 | 385 |
| Positive | 0 | 165 | 165 |
| Total | 384 | 166 | 550 |
| Second Test | | | |
| Negative | 382 | 1 | 383 |
| Positive | 2 | 165 | 167 |
| Total | 384 | 166 | 550 |

**TABLE 5** Intra- and Inter-laboratory agreement of cobas (5800) HPV test for HPV type-specific detection

| HPV result | Intra-laboratory | | Inter-laboratory | |
|---|---|---|---|---|
| | reproducibility (95% CI) | kappa | reproducibility (95% CI) | kappa |
| HPV16 | 99.8% (99.2–99.96%) | 0.981 | 99.6% (98.9–99.88%) | 0.964 |
| HPV18 | 99.8% (99.2–99.96%) | 0.922 | 100% (99.5–100%) | 1.000 |
| ther | 99.8% (99.2–99.96%) | 0.995 | 99.8% (99.2–99.96%) | 0.995 |

provides a highly automated option for low-to-medium volume laboratories, and while it only reports 144 clinical samples in 8 hours, the instrument can be loaded with 240 clinical specimens in 8 hours, which can be run without staff intervention. The cobas 5800 system also has a smaller footprint, which may free up more space within the laboratory.

In conclusion, the cobas HPV test when run on the Roche cobas 5800 system fulfils all requirements, as outlined by an international consortium (7), for being a clinically validated assay that can be used to for primary cervical screening.

## ACKNOWLEDGMENTS

Roche had no role in the study design, data collection, analysis and interpretation of the data, manuscript preparation, and the decision to publish the present manuscript.

M.S. and D.H. are investigators on the Compass Trial for which the ACPCC has received funding from Roche Molecular Systems. Marion Saville is a Chief Investigators on the National Health and Medical Research Council-funded Center for Research Excellence in Cervical Cancer Control (APP1135172), which provides partial salary support to David Hawkes. M.A. was supported by the Horizon 2020 Framework Program for Research and Innovation of the European Commission, through the RISCC Network (Grant No. 847845). Sciensano, the employer of M. Arbyn received funding from the European Society of Gynecological Oncology (ESGO), German Guideline Program in Oncology (German Cancer Aid project), World Health Organization (Geneva, Switzerland, via Agreement for Performance of Work for guidelines on Screening and Treatment of Pre-Invasive Cervical Disease); the European Commission Initiative on Cervical Cancer (EC-CvC) and VALGENT (see Arbyn J Clin Virol 2016).

## AUTHOR AFFILIATIONS

[1]Australian Centre for the Prevention of Cervical Cancer, Carlton, Victoria, Australia
[2]Molecular Laboratory, Vivalia Hospital, Arlon, Belgium
[3]Department of Physiology, Anatomy and Microbiology, LaTrobe University, Bundoora, Victoria, Australia
[4]Belgian Cancer Centre/Unit of Cancer Epidemiology, Sciensano, Brussels, Belgium
[5]Department of Obstetrics and Gynaecology, University of Malaya, Kuala Lumpur, Malaysia
[6]Department of Biochemistry and Pharmacology, The University of Melbourne, Victoria, Australia
[7]Department of Pathology, University of Malaya, Kuala Lumpur, Malaysia

## AUTHOR ORCIDs

Marc Arbyn  http://orcid.org/0000-0001-7807-5908
Marion Saville  http://orcid.org/0000-0002-5924-446X
David Hawkes  http://orcid.org/0000-0002-3882-801X

## FUNDING

| Funder | Grant(s) | Author(s) |
|---|---|---|
| Roche Holding \| Roche Diagnostics (Roche Diagnostics Corporation) | NA | David Hawkes |

## AUTHOR CONTRIBUTIONS

Nikita Mehta, Conceptualization, Data curation, Formal analysis, Funding acquisition, Investigation, Methodology, Project administration, Supervision, Writing – original draft, Writing – review and editing | Marco Ho Ting Keung, Data curation, Funding acquisition, Investigation, Methodology, Project administration, Supervision, Writing – review and editing | Eunice Pineda, Data curation, Formal analysis, Investigation, Methodology, Validation, Writing – review and editing | Elliott Lynn, Investigation, Methodology, Supervision, Writing – review and editing | Dagnachew Fetene, Data curation, Formal analysis, Investigation, Methodology, Project administration, Writing – review and editing | Alvin Lee, Formal analysis, Investigation, Methodology, Project administration, Writing – review and editing | Nicolas Hougardy, Investigation, Methodology, Writing – review and editing | Amelie Heinrichs, Investigation, Methodology, Writing – review and editing | Hiu Tat Mark Chan, Formal analysis, Methodology, Supervision, Writing – review and editing | Marc Arbyn, Data curation, Formal analysis, Methodology, Validation, Writing – review and editing | Marion Saville, Funding acquisition, Investigation, Supervision, Writing – review and editing | David Hawkes, Conceptualization, Data curation, Formal analysis, Funding acquisition, Investigation, Methodology, Project administration, Supervision, Writing – original draft, Writing – review and editing

## ETHICAL APPROVAL

This study was approved by Bellberry Human Research Ethics Committee (2023–11-1356).

## ADDITIONAL FILES

The following material is available online.

### Open Peer Review

**PEER REVIEW HISTORY (review-history.pdf).** An accounting of the reviewer comments and feedback.

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
