## [Reviewer comments · Microbiology Spectrum]

Microbiology Spectrum

Clinical validation of the Roche cobas HPV test on the Roche cobas 5800 system for the purpose of cervical screening

Nikita Mehta, Marco Keung, Eunice Pineda, Elliott Lynn, Dagnachem Fetene, Alvin Lee, Nicolas Hougardy, Amelie Heinrichs, Hiu Tat Chan, Marc Arbyn, Marion Saville, and David Hawkes

Corresponding Author(s): David Hawkes, Australian Centre for the Prevention of Cervical Cancer

Review Timeline:

Submission Date:	June 19, 2024
Editorial Decision:	July 15, 2024
Revision Received:	August 2, 2024
Accepted:	August 9, 2024

Editor: Sophia Georghiou

Reviewer(s): Disclosure of reviewer identity is with reference to reviewer comments included in decision letter(s). The following individuals involved in review of your submission have agreed to reveal their identity: Jesper Bonde (Reviewer #1)

Transaction Report:

DOI: <https://doi.org/10.1128/spectrum.01493-24>

Re: Spectrum01493-24 (Clinical validation of the Roche cobas HPV test on the Roche cobas 5800 system for the purpose of cervical screening)

Dear Prof. David Hawkes:

Thank you for the privilege of reviewing your work. Below you will find my comments, instructions from the Spectrum editorial office, and the reviewer comments.

Revision Guidelines

Sincerely,
Sophia Georghiou
Editor
Microbiology Spectrum

Reviewer #1 (Comments for the Author):

The manuscripts by Mehta et al details the validation of the Roche cobas HPV test on the Roche Cobas5800 instrument platform using the cobas4800 systems as comparator.

The manuscript is an easy read, the methodology is well defined and described and comply with international validation guidelines within the field.

Minor revisions:

Line 55, find the spelling error...

Line 56, write out TPP as it is not previously introduced in the text

Line 62, Regarding reproducibility. No defined timeframe as "several weeks apart" exist, and it is BTW reproducibility, not stability that this criterion is to assess. The Meijer guidelines do not stipulate a specific time frame. Please revise.

Line 71, Suggest to change "clinically validated" to "clinically evaluated" as some dissent exists on whether the two references claiming validation of the cobas 6800 in fact comply with the international guidelines using cobas 4800 as comparator test at a point in time when this assay was not a consensus comparator test.

Line 101, Please specify that the absence of disease in the control group was not verified by colposcopy and biopsies.

Finally, for discussion, the readers would like to know how much more the 5800 turnaround is compared to the 4800.

Reviewer #2 (Comments for the Author):

Clinical validation of the Roche cobas HPV test on the Roche cobas 5800 system for the purpose of cervical screening
Mehta et al.

This short article describes the results of the application of international guidelines ("Meijer criteria") to assess the clinical performance of the cobas 5800 HPV assay and determine its suitability for use in primary HPV screening. I found it to be well written, clear, and concise. As the authors explain, the criteria were originally intended to be assessed using either the Hybrid Capture 2 assay or the GP5+/6+ PCR-based reference standard, which in recent years has evolved to the use of newer clinically validated assays such as cobas 4800, used in this study (GP5+/6+ is not commercially available and Hybrid Capture 2 utilizes a less sensitive signal amplification technology which does not provide any genotyping information). The data presented demonstrates that the cobas 5800 assay readily meets the benchmarks for sensitivity and specificity versus endpoint CIN2+ disease and those for intra- and inter-lab reproducibility (The relative clinical sensitivity was 1.000, when compared with the cobas 4800 HPV test and the relative clinical specificity was 0.995; Intra- and inter-laboratory agreement were 99.1% and 99.6%, respectively). The authors conclude that the cobas 5800 HPV assay is suitable for use in primary HPV screening.

There was some missing contextual information and I have the following clarifying questions / comments for the authors' consideration:

1) The cobas 5800 is a newly released platform, which recently obtained FDA approval for use with the cobas HPV assay. The manuscript would benefit from brief description of the pre-analytic and analytic capabilities of the system, and whether the study tested samples directly transferred from the vial (which per their US PI, requires an open vial and tube transfer) or were processed manually. The authors imply that the 5800 is designed to replace the older 4800 System and that it is faster and more versatile (lines 186-189). However, it appears to offer less throughput than cobas 4800 per the Roche website (144 vs 192 samples in an 8-hour shift).

2) The authors refer to the "Roche cobas HPV test" and discuss the use of 4 systems offered by the company (cobas 4800, 6800, 8800 and the newer cobas 5800 System, described herein). However, there are two versions of the Roche cobas HPV assay currently available - one run on the cobas 4800 System, and a different assay which is run on the cobas 6800/8800 Systems. My understanding is that the two assays have different sample input volumes. The cobas 6800/8800 assay has an earlier cut-off (the clinical cutoff is set at Ct of 38.5 for HPV 16, Ct of 38.0 for HPV 18 and Ct of 34.5 for all other HR HPV genotypes; the cobas 4800 cut-off is 40.5 for HPV16 and 40 for all other HR types). In addition, their respective PIs describe 10% whole blood tolerance for the 6800/880 assay versus a contra-indication for the 4800 version, suggesting an improvement in the extraction chemistry/workflow. From a regulatory and clinical standpoint, they are distinct assays. Thus, a key question is, which version of the assay was validated on the cobas 5800 system?

3) A related point to 2) above would be to include Ct scores when discussing discordant results to clarify whether negative results on one assay were simply beyond the clinical cutoff or were completely negative.

4) The cobas 4800 is FDA-approved and clinically validated but it has been reported to have less agreement among control samples when compared to other FDA-approved assays (likely related to its late Ct clinical cutoffs) 1, further underlining the importance of identifying which assay was used for the study. The authors report (line 72) that the cobas 5800 assay performance will also be compared to 6800 assay performance in a forthcoming yet to be published study.

5) The study design uses best practice sample selection methodologies, utilizing "consecutive" screening samples enriched with those with CIN2+ disease and double negative history (cytology and HPV) for control (< CIN2) samples (line 101). This avoids the introduction of bias by using strong positive and negative samples, which would artificially inflate assay agreement (~90% of

samples have NILM cytology but may be HPV positive which is more likely to challenge the clinical cutoff of the assay). This type of flawed design has actually been used by others to compare cobas 4800 and cobas 6800 performance (employing the exclusive use of CIN3+ positives and double negatives)². It is not completely clear how the reproducibility samples were obtained (lines 113-115): "For intra-laboratory reproducibility, we identified a set of 550 cervical samples, of which 165 (30%) had HPV detected on the cobas (5800)". Please clarify if the negatives used for reproducibility testing were also "consecutive" routine screening samples and if the positives were also randomly selected.

(1) Innamaa, A.; Dudding, N.; Ellis, K.; Crossley, J.; Smith, J. H.; Tidy, J. A.; Palmer, J. E. High-risk HPV platforms and test of cure: should specific HPV platforms more suited to screening in a 'test of cure' scenario be recommended? *Cytopathology* : official journal of the British Society for Clinical Cytology 2015, 26 (6), 381-387. DOI: 10.1111/cyt.12223 From NLM.

(2) Sundström, K.; Lamin, H.; Dillner, J. Validation of the cobas 6800 human papillomavirus test in primary cervical screening. *PloS one* 2021, 16 (2), e0247291. DOI: 10.1371/journal.pone.0247291 From NLM.

Clinical validation of the Roche cobas HPV test on the Roche cobas 5800 system for the purpose of cervical screening

Mehta et al.

This short article describes the results of the application of international guidelines (“Meijer criteria”) to assess the clinical performance of the cobas 5800 HPV assay and determine its suitability for use in primary HPV screening. I found it to be well written, clear, and concise. As the authors explain, the criteria were originally intended to be assessed using either the Hybrid Capture 2 assay or the GP5+/6+ PCR-based reference standard, which in recent years has evolved to the use of newer clinically validated assays such as cobas 4800, used in this study (GP5+/6+ is not commercially available and Hybrid Capture 2 utilizes a less sensitive signal amplification technology which does not provide any genotyping information). The data presented demonstrates that the cobas 5800 assay readily meets the benchmarks for sensitivity and specificity versus endpoint CIN2+ disease and those for intra- and inter-lab reproducibility (The relative clinical sensitivity was 1.000, when compared with the cobas 4800 HPV test and the relative clinical specificity was 0.995; Intra- and inter-laboratory agreement were 99.1% and 99.6%, respectively). The authors conclude that the cobas 5800 HPV assay is suitable for use in primary HPV screening.

There was some missing contextual information and I have the following clarifying questions / comments for the authors’ consideration:

- 1) The cobas 5800 is a newly released platform, which recently obtained FDA approval for use with the cobas HPV assay. The manuscript would benefit from brief description of the pre-analytic and analytic capabilities of the system, and whether the study tested samples directly transferred from the vial (which per their US PI, requires an open vial and tube transfer) or were processed manually. The authors imply that the 5800 is designed to replace the older 4800 System and that it is faster and more versatile (lines 186-189). However, it appears to offer less throughput than cobas 4800 per the Roche website (144 vs 192 samples in an 8-hour shift).
- 2) The authors refer to the “Roche cobas HPV test” and discuss the use of 4 systems offered by the company (cobas 4800, 6800, 8800 and the newer cobas 5800 System, described herein). However, there are two versions of the Roche cobas HPV assay currently available – one run on the cobas 4800 System, and a different assay which is run on the cobas 6800/8800 Systems. My understanding is that the two assays have different sample input volumes. The cobas 6800/8800 assay has an earlier cut-off (the clinical cutoff is set at Ct of 38.5 for HPV 16, Ct of 38.0 for HPV 18 and Ct of 34.5 for all other HR HPV genotypes; the cobas 4800 cut-off is 40.5 for HPV16 and 40 for all other HR types). In addition, their respective PIs describe 10% whole blood tolerance for the 6800/880 assay versus a contra-indication for the 4800 version, suggesting an improvement in the extraction chemistry/workflow. From a regulatory and clinical standpoint, they are distinct assays. Thus, a key question is, which version of the assay was validated on the cobas 5800 system?

- 3) A related point to 2) above would be to include Ct scores when discussing discordant results to clarify whether negative results on one assay were simply beyond the clinical cutoff or were completely negative.
- 4) The cobas 4800 is FDA-approved and clinically validated but it has been reported to have less agreement among control samples when compared to other FDA-approved assays (likely related to its late Ct clinical cutoffs)¹, further underlining the importance of identifying which assay was used for the study. The authors report (line 72) that the cobas 5800 assay performance will also be compared to 6800 assay performance in a forthcoming yet to be published study.
- 5) The study design uses best practice sample selection methodologies, utilizing “consecutive” screening samples enriched with those with CIN2+ disease and double negative history (cytology and HPV) for control (< CIN2) samples (line 101). This avoids the introduction of bias by using strong positive and negative samples, which would artificially inflate assay agreement (~90% of samples have NILM cytology but may be HPV positive which is more likely to challenge the clinical cutoff of the assay). This type of flawed design has actually been used by others to compare cobas 4800 and cobas 6800 performance (employing the exclusive use of CIN3+ positives and double negatives)². It is not completely clear how the reproducibility samples were obtained (lines 113-115): “For intra-laboratory reproducibility, **we identified** a set of 550 cervical samples, of which 165 (30%) had HPV detected on the cobas (5800)”. Please clarify if the negatives used for reproducibility testing were also “consecutive” routine screening samples and if the positives were also randomly selected.

(1) Innamaa, A.; Dudding, N.; Ellis, K.; Crossley, J.; Smith, J. H.; Tidy, J. A.; Palmer, J. E. *High-risk HPV platforms and test of cure: should specific HPV platforms more suited to screening in a 'test of cure' scenario be recommended?* *Cytopathology : official journal of the British Society for Clinical Cytology* **2015**, 26 (6), 381-387. DOI: 10.1111/cyt.12223
From NLM.

(2) Sundström, K.; Lamin, H.; Dillner, J. *Validation of the cobas 6800 human papillomavirus test in primary cervical screening.* *PloS one* **2021**, 16 (2), e0247291. DOI: 10.1371/journal.pone.0247291 From NLM.

Response to Reviewers Comments

Clinical validation of the Roche cobas HPV test on the Roche cobas 5800 system for the purpose of cervical screening

Reviewer #1 (Comments for the Author):

The manuscripts by Mehta et al details the validation of the Roche cobas HPV test on the Roche cobas 5800 instrument platform using the cobas 4800 system as comparator.

The manuscript is an easy read, the methodology is well defined and described and comply with international validation guidelines within the field.

We thank the reviewer for their comments

Minor revisions:

Line 55, find the spelling error...

This correction has been made.

Line 56, write out TPP as it is not previously introduced in the text

This suggestion has been actioned.

Line 62, Regarding reproducibility. No defined timeframe as "several weeks apart" exist, and it is BTW reproducibility, not stability that this criterion is to assess. The Meijer guidelines do not stipulate a specific time frame. Please revise.

The sentence has been edited to quote and cite the wording from the Meijer et al, 2009 manuscript.

“Reproducibility is also assessed with a cohort of at least 500 samples tested twice in the same laboratory (intra-laboratory) at the suggested interval of ‘several weeks later’ (7) and then re-tested in an independent external laboratory (inter-laboratory).”

In the Materials and Methods (Lines 117 – 118) the interval of testing for intra-laboratory is given as a range of 34 – 43 days. On Line 120 the inter-laboratory interval is presented as the range of 58 – 77 days.

Line 71, Suggest to change "clinical validated" to "clinical evaluated" as some dissent exists on whether the two references claiming validation of the cobas 6800 in fact comply with the international guidelines using cobas 4800 as comparator test at a point in time when this assay was not a consensus comparator test.

We thank the reviewer for this comment. There is a manuscript by Arbyn and an international consortium which has been provisionally accepted for publication which identifies ‘second-generation standard comparator HPV assays’, including the Roche cobas 4800 HPV test. We would be happy to cite this manuscript but to avoid not following the journal’s guidelines on referencing we cited a presentation at EUROGIN 2024 (Reference 4) below.

Arbyn M. 2024. New guidelines for HPV test validation and current list of validated HPV tests, EUROGIN, Stockholm, Sweden, 13/03/2024.

We would be happy to cite the full manuscript if the journal is accepting of this.

Line 101, Please specify that the absence of disease in the control group was not verified by colposcopy and biopsies.

This suggestion has been actioned.

Finally, for discussion, the readers would like to know how much more the 5800 turnaround is compared to the 4800.

To clarify the throughput and turnaround of the cobas 5800 system compared to the cobas 4800 system the following text has been modified from Line 194.

“The cobas (5800) is a useful addition as a clinically validated HPV test as it provides a more modern, versatile, and faster (2 h 45 mins to first result) option when compared with the cobas 4800 system (4 h 45 mins to first result) which was launched 15 years ago. The cobas 4800 system is more manual and requires more staff hands on time (estimated as 39 minutes per run of 96 samples. The cobas 5800 system provides a highly automated option for low to medium volume laboratories and whilst it only reports 144 clinical samples in 8 hours, the instrument can be loaded with 240 clinical specimens in 8 hours which can be run without staff intervention. The cobas 5800 system also has a smaller footprint which may free up more space within the laboratory.”

Reviewer #2 (Comments for the Author):

This short article describes the results of the application of international guidelines ("Meijer criteria") to assess the clinical performance of the cobas 5800 HPV assay and determine its suitability for use in primary HPV screening. I found it to be well written, clear, and concise. As the authors explain, the criteria were originally intended to be assessed using either the Hybrid Capture 2 assay or the GP5+/6+ PCR-based reference standard, which in recent years has evolved to the use of newer clinically validated assays such as cobas 4800, used in this study (GP5+/6+ is not commercially available and Hybrid Capture 2 utilizes a less sensitive signal amplification technology which does not provide any genotyping information). The data presented demonstrates that the cobas 5800 assay readily meets the benchmarks for sensitivity and specificity versus endpoint CIN2+ disease and those for intra- and inter-lab reproducibility (The relative clinical sensitivity was 1.000, when compared with the cobas 4800 HPV test and the relative clinical specificity was 0.995; Intra- and inter-laboratory agreement were 99.1% and 99.6%, respectively). The authors conclude that the cobas 5800 HPV assay is suitable for use in primary HPV screening.

We thank the reviewer for these comments.

There was some missing contextual information, and I have the following clarifying questions / comments for the authors' consideration:

1) The cobas 5800 is a newly released platform, which recently obtained FDA approval for use with the cobas HPV assay. The manuscript would benefit from brief description of the pre-analytic and analytic capabilities of the system, and whether the study tested samples directly transferred from the vial (which per their US PI, requires an open vial and tube transfer) or were processed manually. The authors imply that the 5800 is designed to replace the older 4800 System and that it is faster and more versatile (lines 186-189). However, it appears to offer less throughput than cobas 4800 per the Roche website (144 vs 192 samples in an 8-hour shift).

Specimens were processed using the Roche p480 pre-analytic instrument which vortexes the primary ThinPrep vial before uncapping, aliquoting 1 ml into a Roche cobas secondary tube, before recapping the primary ThinPrep vial.

The following wording has been added to the manuscript at Line 95.

“ Cervical specimens in the primary ThinPrep vial were processed using the Roche 480 pre-analytic instrument which aliquots 1 ml of the specimen into a Roche cobas secondary tube which can then be run on both the Roche cobas 4800 and cobas 5800 systems. ”

To clarify the throughput and turnaround of the cobas 5800 system compared to the cobas 4800 system the following text has been modified from Line 194.

“The cobas (5800) is a useful addition as a clinically validated HPV test as it provides a more modern, versatile, and faster (2 h 45 mins to first result) option when compared with the cobas 4800 system (4 h 45 mins to first result) which was launched 15 years ago. The cobas 4800 system is more manual and requires more staff hands on time (estimated as 39 minutes per run of 96 samples. The cobas 5800 system provides a highly automated option for low to medium volume laboratories and whilst it only reports 144 clinical samples in 8 hours, the instrument can be loaded with 240 clinical specimens in 8 hours which can be run without staff intervention. The cobas 5800 system also has a smaller footprint which may free up more space within the laboratory.”

2) The authors refer to the "Roche cobas HPV test" and discuss the use of 4 systems offered by the company (cobas 4800, 6800, 8800 and the newer cobas 5800 System, described herein). However, there are two versions of the Roche cobas HPV assay currently available - one run on the cobas 4800 System, and a different assay which is run on the cobas 6800/8800 Systems. My understanding is that the two assays have different sample input volumes. The cobas 6800/8800 assay has an earlier cut-off (the clinical cutoff is set at Ct of 38.5 for HPV 16, Ct of 38.0 for HPV 18 and Ct of 34.5 for all other HR HPV genotypes; the cobas 4800 cut-off is 40.5 for HPV16 and 40 for all other HR types). In addition, their respective PIs describe 10% whole blood tolerance for the 6800/880 assay versus a contra-indication for the 4800 version, suggesting an improvement in the extraction chemistry/workflow. From a regulatory and clinical standpoint, they are distinct assays. Thus, a key question is, which version of the assay was validated on the cobas 5800 system?

We thank the author for querying the terminology. This has been an ongoing source of confusion as prior to the cobas HPV test being released the cobas 4800 HPV test was often referred to in the literature as the cobas HPV tests (and occasionally still is). We have tried to make sure that we have clarified the terminology as much as possible with the information from the text below.

‘cobas 4800’ is defined on Line 67 of the body text as being the Roche cobas 4800 HPV test processed on the Roche cobas 4800 system.

Line 70 states that *“The Roche cobas HPV test, which can be processed on the Roche 5800, 6800, and 8800 systems”*

‘cobas (5800)’ is defined on Line 75 of the body text as being the Roche cobas HPV test processed on the Roche cobas 5800 system.

As the reviewer correctly identifies, the cobas 4800, and cobas HPV tests are different including in their chemistry, analytical sensitivity and a range of other aspects. Identifying which of these two assays has been used is critical in understanding the value of an HPV result.

3) A related point to 2) above would be to include Ct scores when discussing discordant results to clarify whether negative results on one assay were simply beyond the clinical cutoff or were completely negative.

The following changes have been made to the text to identify where discordant samples at or beyond the manufacturers stated limit of detection, rather than individual ct values.

Line 149

“Two samples were discordant with one sample, a CIN2 case, for which HPV was not detected on the cobas 4800 but HPV Other was detected on the cobas (5800) with a ct value indicating a moderate positive result (> 6 cycle threshold (ct) stronger than manufacturer’s stated limit of detection (LOD) ct value)(Table 2). The second discordant sample was a CIN3 case that had HPV Other detected (beyond LOD ct) on the cobas 4800, but no HPV detected on the cobas (5800).”

Line 160

“Of the five controls that had HPV detected on the cobas (5800) HPV test but were negative on the cobas 4800, HPV 16 was detected in four, and HPV Other in the other specimen. All five results were at or beyond the LOD ct value.”

Line 171

“Discordant results: Of the three discordant specimens, all tested negative in the first intra-laboratory test. Two specimens only detected HPV for the second test, one detected HPV16 and the other detected HPV Other. One specimen only detected HPV (Other) for the intra-laboratory (Vivalia) test. All three discordant specimens had results at or beyond the LOD ct values.”

4) The cobas 4800 is FDA-approved and clinically validated but it has been reported to have less agreement among control samples when compared to other FDA-approved assays (likely related to its late Ct clinical cutoffs) 1, further underlining the importance of identifying which assay was used for the study. The authors report (line 72) that the cobas 5800 assay performance will also be compared to 6800 assay performance in a forthcoming yet to be published study.

We thank the reviewer for their feedback. We hope the information above has clarified exactly which assay is being used at all points within the manuscript. We have removed reference to other studies in the manuscript.

5) The study design uses best practice sample selection methodologies, utilizing "consecutive" screening samples enriched with those with CIN2+ disease and double negative history (cytology and HPV) for control (< CIN2) samples (line 101). This avoids the introduction of bias by using strong positive and negative samples, which would artificially inflate assay agreement (~90% of samples have NILM cytology but may be HPV positive which is more likely to challenge the clinical cutoff of the assay). This type of flawed design has actually been used by others to compare cobas 4800 and cobas 6800 performance (employing the exclusive use of CIN3+ positives and double negatives)². It is not completely clear how the reproducibility samples were obtained (lines 113-115): "For intra-laboratory reproducibility, we identified a set of 550 cervical samples, of which 165 (30%) had HPV detected on the cobas (5800)". Please clarify if the negatives used for reproducibility testing were also "consecutive" routine screening samples and if the positives were also randomly selected.

The following statement has been added from Line 117 to clarify that the same method for the specificity cohort was used for the reproducibility cohort.

"For intra-laboratory reproducibility, we identified a set of 550 cervical samples, which were identified in the same manner as the specificity cohort described above until there were 385 HPV not detected and 165 HPV detected (30%) on the cobas (5800)."

Peer Reviewer References

(1) Innamaa, A.; Dudding, N.; Ellis, K.; Crossley, J.; Smith, J. H.; Tidy, J. A.; Palmer, J. E. High-risk HPV platforms and test of cure: should specific HPV platforms more suited to screening in a 'test of cure' scenario be recommended? *Cytopathology : official journal of the British Society for Clinical Cytology* 2015, 26 (6), 381-387. DOI: 10.1111/cyt.12223 From NLM.

(2) Sundström, K.; Lamin, H.; Dillner, J. Validation of the cobas 6800 human papillomavirus test in primary cervical screening. *PloS one* 2021, 16 (2), e0247291. DOI: 10.1371/journal.pone.0247291 From NLM.

Re: Spectrum01493-24R1 (Clinical validation of the Roche cobas HPV test on the Roche cobas 5800 system for the purpose of cervical screening)

Dear Prof. David Hawkes:

Your manuscript has been accepted, and I am forwarding it to the ASM production staff for publication. Your paper will first be checked to make sure all elements meet the technical requirements. ASM staff will contact you if anything needs to be revised before copyediting and production can begin. Otherwise, you will be notified when your proofs are ready to be viewed.

Sincerely,
Sophia Georghiou
Editor
Microbiology Spectrum